# Analysis of Land-Use Change in Shortandy District in Terms of Sustainable Development

**Onggarbek Alipbeki** [1,*] , **Chaimgul Alipbekova** [2] , **Arnold Sterenharz** [3] , **Zhanat Toleubekova** [1] , **Saule Makenova** [1] , **Meirzhan Aliyev** [1] **and Nursultan Mineyev** [1]

1   Department of Land Use and Geodesy, S.Seifullin Kazakh Agro Technical University, Jhenis Avenue, 62, Nur-Sultan 010011, Kazakhstan; jtoleubekova@mail.ru (Z.T.); saule_makenova@mail.ru (S.M.); meirzhan.maratuly@mail.ru (M.A.); nursultan_23@list.ru (N.M.)
2   Department of Plant Protection and Quarantine, S.Seifullin Kazakh Agro Technical University, Jhenis Avenue, 62, Nur-Sultan 010011, Kazakhstan; chaimgul@mail.ru
3   EXOLAUNCH GmbH, (Spin-off Company from the Technical University of Berlin, Germany), Reuchlin str. 10, 10553 Berlin, Germany; arnold-com@yandex.ru
*   Correspondence: oalipbeki@mail.ru; Tel.: +7-7715369615

**Abstract:** The suburban territories of large cities are transitional zones where intensive transformations in land use are constantly taking place. Therefore, the presented work is devoted to an integrated assessment of land use changes in the Shortandy district (Kazakhstan) based on an integrated study of the dynamics of land use and sustainable development indicators (SDIs). It was found that the main tendency in the land use of this Peri-urban area (PUA) during 1992–2018 is their intensification, through an increase in arable lands. Kazakhstan only recently started the systematic collection of SDIs according to international standards. Therefore, to assess the sustainable development of the study area, limited amounts of information were available. Nevertheless, the use of SDIs from 2007 to 2017 showed that the growth of economic development inthe study area is almost adequately accompanied by an increase in the level of social and environmental development. The methodological approach used can be widely used to assess the sustainable development of specific territories in general and the development of the capital of Kazakhstan and their PUA, in particular.

**Keywords:** land use change; analyze; sustainable development; Shortandy district

## 1. Introduction

According to the Food and Agriculture Organization (FAO) statement on food security "by 2050, the world's population will grow to almost 10 billion," which will increase food demand by about 50% compared to 2013 [1]. At the same time, the share of the rural population will decrease, and the urban population of the world will reach 68% [2]. Increased food production is recommended to be accompanied by sustainable agricultural land management [3], including PUA.

The purpose of increasing the effectiveness of land use management is to stop or at least slow down the negative impact of land use on natural resources. Moreover, adverse processes are often understood as degradation of the soil cover under the influence of various types of erosion [4], desertification and salinization [5,6], depletion of soil fertility [7], pollution [8], reduced water quality [9], land grabs by rapidly growing cities and their consequences [10], etc. These local changes in land use together have a global impact on climate, hydrology, biogeochemistry, biodiversity and the ability of biological systems to meet human needs [11,12]. Besides, changes in land use significantly affect the energy balance of the entire Earth and the biogeochemical cycles in it, of which 60% are associated with direct human activities (for example, urban sprawl and intensification of agriculture) and only 40% with

indirect environmental factors (for example, climate change) [13–17].Ultimately, these undesirable processes occurring in the environment, if detected and not prevented in time, lead to undesirable economic, social and environmental consequences, the indicators of which should also be measured and evaluated. It is emphasized that the process of sustainable development is multidimensional and interdisciplinary, and the indicators proposed for its assessment are an attempt to combine them into a measurable set [18–21], which usually focus on a certain aspect of sustainable development.

In the light of the above context, there is an urgent need for a comprehensive and systematic assessment of changes in land use, environmental factors, economic and social conditions by instrumental and statistical methods based on indicators of sustainable development (SDI) [22].

On the one hand, the transformation of land use is the main driver of environmental change at all levels. Instrumental methods using Remote Sensing (RS) and Geographic Information Systems (GIS) are widely used to evaluate Land Use and Land Cover (LULC) changes. The accuracy of the LULC assessment using RS and GIS depends on the potential of the devices used and their sensors, the frequency of measurement repeatability and the qualifications of an expert [23–25]. For example, the assessment of long-term LULC changes based on RS and GIS is implemented at the global, regional, national and local levels [26–31]. It is obvious that over time, the level of reliability of the assessment of changes in LULC will increase with the development of geoinformatics, in general, and geoinformation technologies, in particular. Apparently, instrumental research methods supplemented by data on the state of social, economic and environmental factors may be more useful in assessing the sustainable development of a particular territory.

On the other hand, the concept of sustainable development is one of the doctrines of the economy and assumes that "it meets the needs of the present, without compromising the ability of future generations to satisfy their own needs" [32]. The Sustainable Development Goals (SDGs) are the foundation for a better and sustainable future for all. They address the global challenges that we face, including those related to poverty, inequality, climate change, environmental degradation, peace and justice. All 17 SDGs are interlinked [33] and provide for the balance between economic growth (economic aspect), care for nature (environmental aspect) and quality of life (social aspect), and are closely related to space and land use. To assess the transformation of sustainable development, different approaches are used [34]. In our opinion, studies on sustainable development using LULC digital maps in integration with economic social and environmental indicators are of the greatest interest. This is because land use always represents a correlation between different economic, social and environmental needs [35]. For example, LULC digital maps were used in conjunction with environmental statistics [35,36], land use intensity [37], socio-economic consequences of land use [38], and a combination of environmental, economic and social factors based on land processing [22]. In order to conduct such comprehensive studies, in addition to reliable digital cards, reliable SDI [39,40] and adequate methodological approaches are additionally needed.

In Kazakhstan, to date, research in the field of sustainable development has been carried out either using LULC digital maps [41], or using reliable scientific and official source statistical data [42]. Initial statistical indicators of sustainable development to date in the republic have not been fully systematized according to the requirements of the SDGs, the use of which still requires their transformation into three or more stages. Nevertheless, the republic fully supports the principles of sustainable development [43] and the country has joined the United Nations (UN) special program "Sustainable Development Goals for the Period until 2030" [44]. Therefore, to monitor the sustainable development of land use, the RS group was created from KazEOSAT 1 and KazEOSAT 2 (Kazakhstan) [45]. In addition, in recent years, studies have been launched in the field of long-term observation of changes in LULC using RS [41,46–51]. However, the problem of a comprehensive assessment of the sustainable development of PUA using instrumental methods for studying changes in land use in combination with the use of SDI remains open.

Based on the foregoing, the goal of our research is a comprehensive assessment of the sustainable development of the Shortandy region, which is the PUA of the metropolis Nur-Sultan, using

instrumental and statistical indicators of sustainable development. The research objectives are the development of spatial and temporal LULC maps for determining changes in land use trends, as well as assessing the level of sustainable development of the Shortandy district based on a multi-step transformation of the currently available initial statistical indicators of sustainable development in the fields of ecology, economics and social conditions.

## 2. Materials and Methods

### 2.1. Study Area

The research area is the Shortandy district, Akmola oblast, which is located on the northern border of the city Nur-Sultan which isthe capital of Kazakhstan (Figure 1), where lack of free space is a problem, as in many other metropolises.

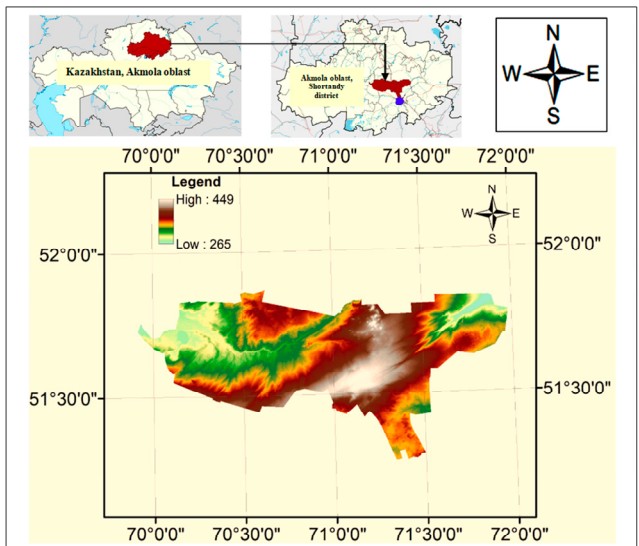

**Figure 1.** Digital elevation map [52] of the Shortandy district.

The area of interest (AOI) covers an area of 4675.6 km$^2$. There are 11 villages in the district. The population of the district as of January 1, 2018, was 29,421 people. The region specializes in gold mining, grain production, livestock farming and processing of agricultural products. The industry focus is agrarian-industrial. A railway passes through the territory of the Shortandy region in several directions: Almaty-Petropavlovsk, Kokshetau-Kyzylorda, etc., roads of international, republican and regional significance, which makes it attractive both for the development of industry and agriculture. The hydrographic network is represented by 11 lakes and several small drains, the flow of which is insignificant. The main water artery flowing through the territory of the district is the Damsa River [53–55].

### 2.2. Data

Landsat data 5 and 8 [56] for 1992, 1998, 2008, 2018 were used to study land use changes in the AOI. The Metadata of images are: LT51550241992155ISP00; LT51560241992162ISP00; LT51550241998267BIK00; LT51560241998258BIK00; LT51550242008103BJC01; LT51560242008126KHC01; LC81550242018146LGN00; LC81560242018185LGN00.

To assess the sustainability of the development of the district, we used statistical data obtained from the relevant internet resources of Kazakhstan [57,58], as well as from the official data provided by the "Republican Scientific and Methodological Centre of the Agrochemical Service" of the Ministry of Agriculture of the Republic of Kazakhstan" (RSMCAS).

### 2.3. Methods

Methodology for classifying land use, it's accuracy assessment and land use map generation is described in our previous work [41], which used the methodological approaches and solutions given in [52,56,59–62]. Methods of assessing sustainable development include four steps (Figure 2) [63].

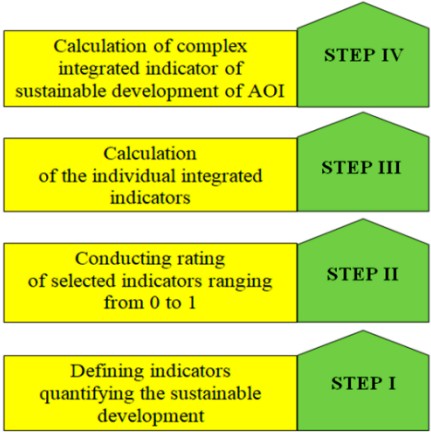

**Figure 2.** Methods of assessing sustainable development [63] of Shortandy district.

Table 1 shows the three groups of SDIs we used: economic, social, and environmental.

**Table 1.** Indicators for the assessment of sustainable development of Shortandy district.

| Indicators | Indicator's Title | Unit of Measurement |
|---|---|---|
| Economic | The volume of industrial output | million tenge |
| | The volume of output of plant products | million tenge |
| | The volume of output of animal products | million tenge |
| | Investment in fixed capital | million tenge |
| Social | Population; | thousand people |
| | Ratio of the average monthly salary to the average wage in the economy as a hole | % |
| | Unemployment rate; | % |
| | Natural population migration coefficient | % |
| | Provision of rural population with drinking water | % |
| | The share of paved roads in the total length of roads | Per mille |
| Environmental | The weighted average content of humus; | % |
| | The weighted average content of easily mobile nitrogen | % |
| | The weighted average content of phosphorus; | % |
| | The weighted average content of exchangeable potassium | % |

## 3. Results

### 3.1. Land Use Changes in Shortandy District

The land use classification of the Shortandy district for the period 1992–2018 indicates the presence of noticeable changes in land use (Tables 2 and 3). Agricultural land occupies the bulk of the AOI (~96%, with arable land ~66% and pasture ~30%), which is clearly seen in Figure 3. In 1992, arable lands and pastures amounted to 95.83%; in 1998, 95.84%; in 2008, 95.71%; and in 2018, 95.61% (Table 2). From the above data, it can be seen that there was a gradual increase in the area of arable land mainly

due to pasture ploughing (Table 3). From 1992 to 1998, there was only a slight tendency to reduce pastures and increase arable lands. Noticeable increases in the share of arable land began in 1998. The territories occupied by arable land from 1992 to 2008 increased by 1.4 km$^2$, and from 1992 to 2018 by 16.5 km$^2$.

**Table 2.** Characteristics of land use changes in Shortandy district.

| Land Use Classes | Area | | | | | | | |
|---|---|---|---|---|---|---|---|---|
| | 1992 | | 1998 | | 2008 | | 2018 | |
| | km$^2$ | % | km$^2$ | % | km$^2$ | % | km$^2$ | % |
| Arable land | 3057.5 | 66.05 | 3057.70 | 66.06 | 3058.90 | 66.08 | 3074.00 | 66.41 |
| Pasture | 1378.50 | 29.78 | 1378.30 | 29.78 | 1374.40 | 29.69 | 1358.00 | 29.34 |
| Water | 125.00 | 2.70 | 125.00 | 2.70 | 125.00 | 2.70 | 125.00 | 2.70 |
| Forest | 36.70 | 0.79 | 36.70 | 0.79 | 37.40 | 0.81 | 37.60 | 0.81 |
| Built-up area | 31.10 | 0.67 | 31.10 | 0.67 | 33.10 | 0.72 | 34.20 | 0.74 |
| Total | 4628.80 | 100 | 4628.80 | 100 | 4628.80 | 100 | 4628.80 | 100 |
| Overall accuracy (%) | 93.1 | | 92.2 | | 94.7 | | 94.0 | |
| Kappa | 0.85 | | 0.83 | | 0.89 | | 0.89 | |

**Table 3.** Land use area difference of Shortandy district between 1992–2018.

| Land Use Classes | Area of Difference (km$^2$) | | |
|---|---|---|---|
| | 1992–1998 | 1992–2008 | 1992–2018 |
| Arable land | 0.20 | 1.40 | 16.50 |
| Pasture | −0.20 | −4.10 | −20.50 |
| Water | 0.00 | 0.00 | 0.00 |
| Forest | 0.00 | 0.70 | 0.90 |
| Built-up area | 0.00 | 2.00 | 3.10 |

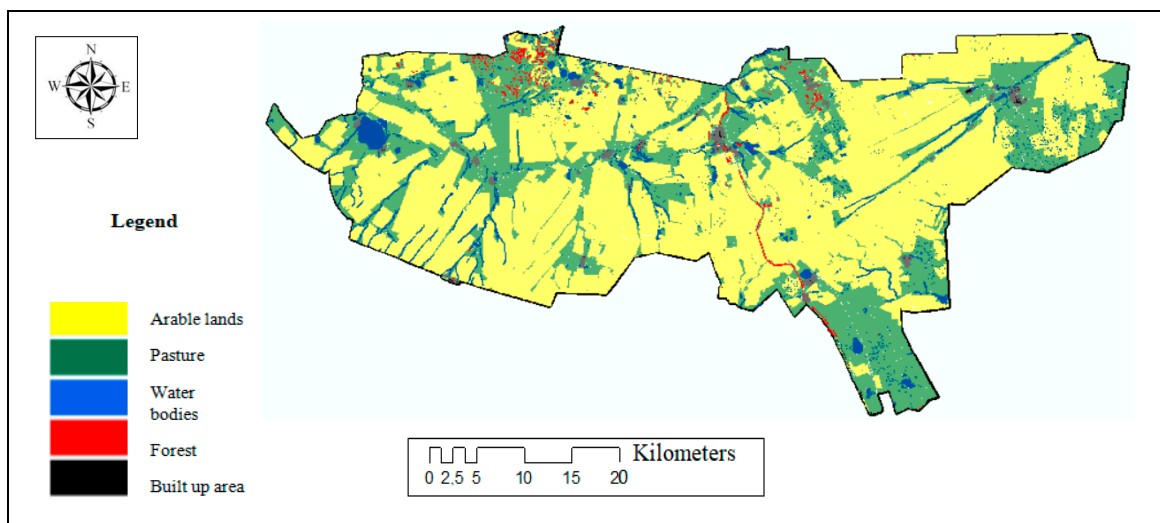

**Figure 3.** Land use map of Shortandy district.

Over the same period (1992–2018), the rangelands AOI decreased by 20.5 km$^2$, of which 16.5 km$^2$ became arable land. A typical example of the expansion of the sown area due to the ploughing of pastures is shown in Figure 4.

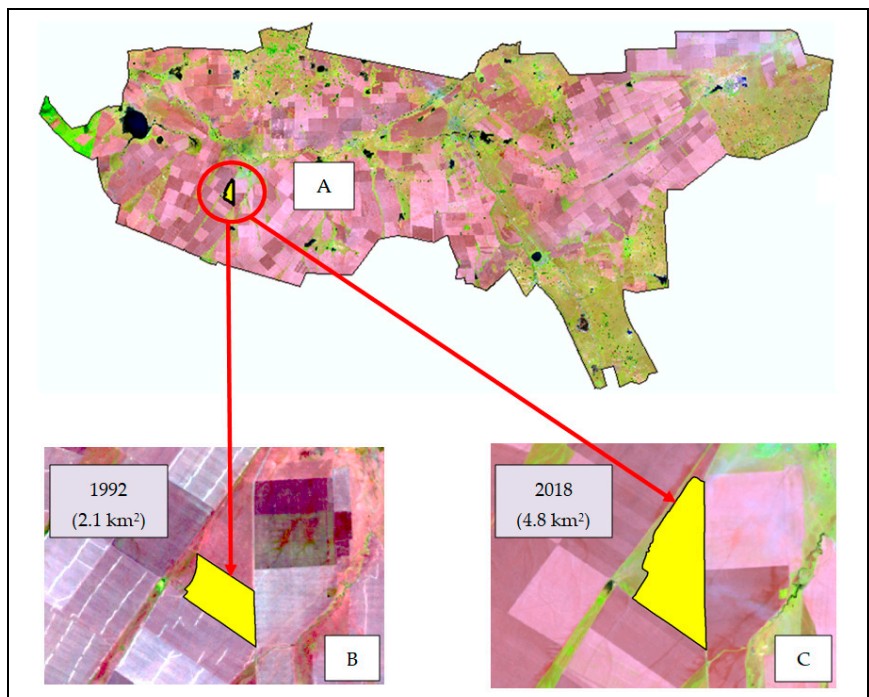

**Figure 4.** An example of change the area of arable land in Shortandy district (**A**) from1992 (**B**) to 2018 (**C**).

The area of water bodies over the years of research remained virtually unchanged and remained at the level of 2.7%.

Forests in the study area occupy less than one percent (0.79%–0.81%). It was noted that the area used for growing trees markedly increased, mainly due to the planting of new forest stands [64].

Urban areas occupy only 0.67%–0.72% of the entire territory of the district. From 1992 to 1998, until the city of Akmola was declared the capital of the republic, the area of the urbanized territories of the district remained unchanged. From 1998 to 2008, the built-up area of the district increased by 2.0 km$^2$. From 1992 to 2018, the total built-up area increased by 3.1 km$^2$ compared to the beginning of our observations.

The overall classification accuracy of land use varied between 92.2%–95.0%. The Kappa coefficient for classified images in 1992 was 0.85; in 1998, 0.83; in 2008, 0.89; and in 2018, 0.89, which indicates the reliability of our land use classification (Table 2).

### 3.2. Analyze of Sustainable Development of Shortandy District

The results of the calculation of SDI are shown in Table 4. In general, there are positive changes intheeconomicandsocialSDI, whichis possible due to strong growth offixed assets in agriculture and industrial production of Shortandy district.

**Table 4.** The individual integrated sustainable development indicators (SDI)of Shortandy district in 2007–2017.

| Year | Economic | Social | Environmental |
|------|----------|--------|---------------|
| 2007 | 0.56 | 0.64 | 0.66 |
| 2008 | 0.58 | 0.74 | 0.69 |
| 2010 | 0.69 | 0.78 | 0.71 |
| 2014 | 0.83 | 0.77 | 0.85 |
| 2017 | 0.91 | 0.77 | 0.85 |

The limitation of the period of assessment of sustainable development from 2007 to 2017 is due to the lack of SDIs that have been conducted hitherto unsystematically on the scale of not only the Shortandy district but the whole republic [65].

In the social sphere, individual indicators are also improving. However, their pace of development is slightly lower than the economic sector. The best SDI in the social sphere was achieved in 2014, after which stagnation was observed.

It should be noted that, according to RSMCAS, an increase in the amountof arable lands is observed, where there is no restoration of soil humus. For example, the weighted average humus content in AOI soils decreased by about 30% compared with 1989, and this process has not completely stopped. At the same time, in recent years, there has been a tendency to increase in the study area soils the mobile form of nitrogen, phosphorus and potassium, which is apparently due to the intensive use of arable land, where it is difficult to obtain high yields without fertilizing.

Evaluation of individual SDI study area allows building hypothetical sustainability testing grounds based on local criteria over some of the years (Figure 5).

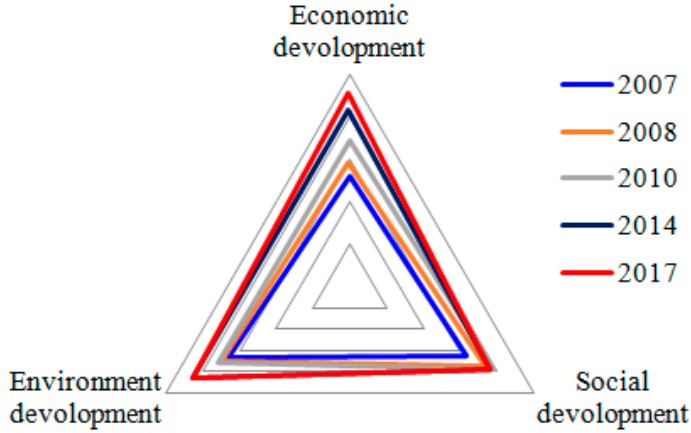

**Figure 5.** Polygons of sustainable development of Shortandy district in 2007–2017.

In general, the nature of the change in the size and shape of landfills convincingly indicates a steady increase in integrated economic, social and environmental indicators. Forms of test polygons are quite smooth, but not always an ideal triangle. This indicates an uneven change in one or another integral indicator over years or measured periods of time, which is quite logical. In this regard, a relatively small deviation of the triangle of 2007 and 2017 towards environmental indicators can be noted, which indicates a noticeable criticality of this indicator in comparison with the economic and social characteristics of the sustainable development of the Shortandy district.

A comprehensive integral indicator of sustainable development of AOI is shown in Figure 6.

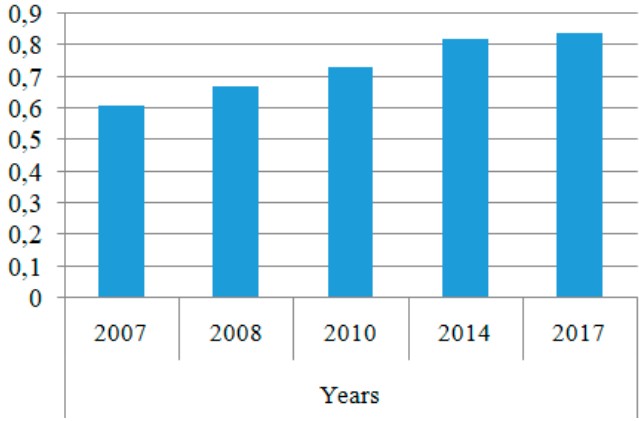

**Figure 6.** The dynamics of the complex SDI of Shortandy district in 2007–2017.

Changes in the comprehensive indicator of sustainable development, which combines all three integrated indicators of economic, environmental and social development, indicate a positive trend. During the estimated period, the process of improving all three sides of development was generally going on, although these dynamics slowed down in the period from 2014 to 2017.

The analysis allows us to conclude about the positive dynamics of the integral index of stability of the Shortandy district. Such changes are primarily associated with a relatively high level of investment in the economy of the study area.

## 4. Discussion

Comprehensive studies aimed at assessing the sustainable development of PUA, which can be divided into two large groups. The first group is land use research using the advantages of instrumental methods (RS and GIS), the second is the widespread use of knowledge-based on transformed statistics in the field of economic and social sciences, as well as environmental protection.

To evaluate changes in land use, RS is presented as a tool to obtain information about an object or a phenomenon at a distance and in a non-destructive way to conduct a spatiotemporal analysis of long-term trends in land use development [22,66]. However, not one of the land use features is measured directly using RS instruments. The relationship between what is measured (radiation) and the characteristics of the land use must be modelled to deduce the last from the first. Therefore, the use of RS to study land usechanges is always accompanied by an assessment of classification accuracy [67].

Interest in the instrumental assessment of land use is very high, since it has moved to a new level [29] and, due to its unique characteristics, can be extremely useful for assessing the sustainable development of territories: local, national, regional and global.

One example in this regard is the Polish Coordination of Information of Environment (CORINE) Land Cover, which assessesthe period from 1990 to 2018 [68]. Interpreting changes in the time horizon, one can obtain information on change trends, which is a valuable guide for the further development of sustainable development policies [22]. This is evidenced by the increase in the depth of land use analysis, associated with sustainable development goals [31,69]. Of considerable interest is the study of processes and determination of sustainable development paths in PUAs, which are caused by modern trends in the growth of urban population in the world [2]. Researches by instrumental methods of specific PUAs are carried out from the following positions: assessing the risk of competition for land-based on spatial indicators [70] the impact of urban expansion on the intensity of agricultural land use [71] and their losses [72,73]; urban planning and management policies [74]; valuation of ecosystem services [75]; assessment of degradation and loss of productive agricultural land [76]; search for driving forces affecting land use [77], etc. Those instrumental approaches to land use assessment can identify the main trends in spatial changes in PUAs for making objective decisions on sustainable development of PUAs.

At the same time, without the use of sustainable development indicators (SDI), based on the transformation of the initial statistical data in the field of economic and social sciences, as well as environmental protection, it is impossible to objectively assess the sustainability of rural development [63], including PUAs.

To this end, a single SDI metadata catalogue and international guidelines have been developed [78]. SDIs have many functions and can lead to more effective decisions by simplifying, refining and making summary information available to politicians. SDIs can help measure and calibrate progress toward sustainable development goals, and can also provide early warning to prevent economic, social and environmental failures [79].

The concept of sustainable development is an attempt to combine growing concern about a number of environmental problems with socio-economic problems [80], which are difficult to accomplish using only instrumental methods. The science of sustainability is based on the study of interdisciplinary connections and combines natural, social, humanitarian, engineering and other sciences to assess the long-term integrity of the environment [81]. For example, in order to identify

mechanisms for sustainable development of PUAs, the processes driving the current global land grabbing are analyzed [82]. The expansion of cities to arable land may be accompanied by a decrease in the sustainability of the development of PUAs [83]; therefore, this problem becomes one of the key research areas, as it is associated with food security [84]. It is argued that a general agricultural and/or socio-economic profile may not be sufficient to understand sustainable development between urban and rural areas and suggest stricter definitions [85], as well as a new approach [86] aimed at identifying the socio-economic consequences of this process. Diversification in suburban agriculture [87], as well as an approach based on smart specialization [88], etc., can play a positive role in increasing the sustainability of the development of PUAs.

Researchers studying the problems of sustainable development of rural areas of Kazakhstan so far consider solutions to this problem at the level of the whole country. For sustainable development, countries propose diversification of the economy [42]; the development of "clean" production, the rational use of natural resources with the maximum possible preservation of the environment through improved technologies [37]; the development of industrial and social infrastructure [89]; solving the problem of accessibility and data quality [90]; studying the positive foreign practice of regulating land relations [91]; and taking into account economic, social, environmental and institutional factors of each region of the republic and choose adequate indicators [92].

For the quantitative assessment of sustainable rural development using indicators of sustainable development, two main approaches are distinguished [93–95]: the creation of separate indicators combined into a system [93,94] and a single integrated indicator [92].

In this regard, international guidelines will serve for national SDI kits, which should be developed taking into account the availability of relevant statistics and reflect the specific situation in countries and specific administrative-territorial units of the country. Therefore, Kazakhstan joined the development of initiatives and measures for sustainable development goals (SDGs) within the framework of the 2030 Agenda [43] and began to collect information according to sustainable development goals indicators [95].

At the same time, the historical imbalance, when a country consumes resources disproportionately compared to their production, is the basis for future problems of sustainable development of Kazakhstan. Calculations showed that reaching the trajectory of "sustainable development" can be ensured if the coefficient of resource utilization is 53%, but not lower than 43% [96]. When forecasting sustainable socio-economic growth, Kazakhstan adheres to three scenarios: optimistic, basic and pessimistic [97]. The forecasted values of Kazakhstan's sustainable growth for 2020–2024, when estimated according to the basic scenario, assumed an oil price of $55 [98]. Real Gross Domestic Product (GDP) growth was projected at 4.1% in 2020. In 2024, it was supposed to reach 4.7%. For five years, the average annual GDP growth rate would be 4.4%. The pessimistic forecast [93] is associated with a decrease in oil prices, which are formed on world markets [99], and the COVID-19 virus epidemic has also added to it [100]. During this period, the government of Kazakhstan is considering the worst option for socio-economic development [101]; the results of such a forecast are not yet available to us. It should be emphasized that in the case of a pessimistic scenario, the adoption of anti-crisis measures is envisaged [97]. They cover measures to ensure macroeconomic stability, including monetary policy instruments, targeted measures to support the real economy, small and medium-sized businesses, and social security. At the same time, depending on the specifics of the crisis, the measures will be revised and adapted to current realities and sustainable development will continue, but its pace will decrease.

The forecast for sustainable development in Russia and Central Asian countries for the future is being formed as in Kazakhstan [102,103]. That is, the sustainable development of Kazakhstan's closest neighbours also depends on the prices of world markets.

It is quite interesting to consider the comparative aspects of the official sustainable development index between Kazakhstan and the Russian Federation [104]. During 1990 and 2015, the sustainable development index in Russia was constantly higher than in Kazakhstan. This indicates the need for

close attention of the government of Kazakhstan to the problem of sustainable development in its republic since the country has already joined the goals of sustainable development 2030.

Thus, the presented material shows that for the most objective assessment of the sustainability of the development of PUAs, it is necessary to use an integrated assessment using instrumental studies of multi-temporal LULC changes and SDI statistical indicators. An example of such an approach already exists [22,35,36,38,105]. However, most of these studies are related to the study of urbanization of cities, and the ways of integrating the results of the LULC study with all three SDI groups (economic, environmental andsocial) for rural areas remain insufficiently studied.In this paper, we also tried to supplement the dynamics of spatiotemporal changes in LULC with SDI analysis using the example of the Shortandy district. In general, the results show the usefulness of the chosen approach, where the development trend of land use and the degree of PUA stability are comprehensively determined. At the same time, due to the limited information on SDI that Kazakhstan has just begun collecting, our work should be considered as an initial step in the chosen areas of research. Nevertheless, the results obtained are of significant value for local and republican bodies interested in developing sustainable development plans.

## 5. Conclusions

As a result of the studies, a comprehensive assessment of the development of the Shortandy region, which is the PUA of a fast-growing metropolis, was carried out. As a result, information was received:

- on the spatiotemporal change in the structure of the LULC using the instrumental analysis method (RS and GIS);
- on the development of the economic, environmental and social potential of the AOI with the use of statistical indicators transformed and combined into three target groups of sustainable development indicators; specifically, economic, ecological and social characteristics.

The study of changes in the land use structure in the AOI from 1992 to 2018 using digital maps revealed an intensification of land use in the study area due to the constant increase in the share of arable land in the LULC structure.

Using the methodology of multi-step conversion of source statistical data into individual, integrated and aggregated of sustainable development indicators revealed that in the last 10years (from 2007 to 2017) there has been a steady development of AOI.

Thus, we have shown that the integrated use of instrumental data and systematic statistical indicators allows us to assess the tendency of land use and the sustainability of the development of a particular agricultural region as a whole. At the same time, due to a lack of initial statistical indicators for AOI, we were not able to evaluate the entire study period covered by the land use study (1992–2018). As a result, we should have limited ourselves to SDI analysis only from 2007 to 2017 with a relatively small number of indicators. Nevertheless, the information obtained in our work is valuable material for interested authorities to plan their activities in the field of sustainable development of a specific PUA. In addition, our approach gives other researchers the opportunity to expand their research in the field of assessing the sustainable development of specific territories, such as the Shortandy district.

The future problems of sustainable development of Kazakhstan are based on the historical imbalance when a country consumes resources disproportionately compared to their production. When forecasting sustainable socio-economic growth, Kazakhstan adheres to three scenarios: optimistic, basic and pessimistic. In all scenarios, sustainable development will continue, but its pace will vary depending on the specifics of the crisis.

**Author Contributions:** Conceptualization, O.A.; methodology, O.A., C.A.; validation, A.S.; formal analysis, A.S., Z.T., S.M.; investigation, M.A., N.M.; writing—original draft preparation, O.A., C.A.; visualization, M.A., N.M.; supervision, O.A.; project administration, O.A. All authors have read and agreed to the published version of the manuscript.

**Funding:** This research was funded by The Committee of Science of the Ministry of Education and Science of the Republic of Kazakhstan under grant number 242 of 03/27/2018 and The ERASMUS+ Programme of the European

Union within the framework of the Project "New and Innovative Courses for Precision Agriculture" (NICOPA) (project reference number 597985-EPP-1-2018-1-KZ-EPPKA2-CBHE-JP). However, this document reflects the views only of the authors, and the Commission cannot be held responsible for any use that may be made of the information contained herein.

**Acknowledgments:** We express our sincere gratitude to the collective of "The Republican Scientific and Methodological Centre of the Agrochemical Service" of the Ministry of Agriculture of the Republic of Kazakhstan for providing information on soil indicators of the Shortandy district from 2007 to 2019.

**Conflicts of Interest:** The authors declare no conflict of interest.

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
