# Peer review of "Analysis of Land-Use Change in Shortandy District in Terms of Sustainable Development"

_land, doi:10.3390/land9050147_

Round 1

Reviewer 1 Report

The paper is improved much. But there are still lots of weaknesses.

  1. Many papers of LAND suggest literature review part. I think your paper needs literature reviews, because framework of indicators is not explained well.
  2. I still cannot find difference of your paper with existing papers. There are many papers about LULC using RS and SDI. Please explain background and purpose of the study in detail.
  3. How about combine the results of LULC and SDI. Detailed assessment by LULC could be the clue for your paper’s difference with existing papers.
  4. Figure 5 is hard to read. And it is overlapped with table 4. Change the figure 5 in detail or delete it.
  5. I cannot find the difference with existing studies in discussion section. What is the improvement in your study area by your paper?

Reviewer 2 Report

Article review: Analyze of Land-Use Change in Shortandy district in Terms of Sustainable Development. Manuscript ID: land-792386
Review of the article after the changes introduced by the authors
The aim of study was an integrated assessment of the land use of the Shortandy district, which is the PUA of the metropolis of Nur Sultan, for sustainable development. The research objectives are the study of the spatiotemporal dynamics of the land use study area and the assessment of the level of sustainable development based on the currently available SDI. The research methodology is based on the research described in the article: "A Spatiotemporal Assessment of Land Use and Land Cover Changes in Peri-Urban Areas: A Case Study of Arshaly District, Kazakhstan" Sustainability, 2020, Volume 12, Issue 4, 1556
Article rating after the changes introduced by the authors:
In its current form (after changes) the article has significantly increased quality.
The authors comprehensively replied to the comments from review 1.
The authors also followed these comments and corrected the text of the article.
However, I think (like the authors) that the methodology should be described. Stages of the study are in the answer to review 1 (it is enough to quote the description of the methodology in points, except for the quote [30]).
Part of the text from point VIII of the answer (VIII.Lack of free space is a common problem in cities ...) should be included in the text - it concerns the characteristics of the research area (The studied agricultural region ...). At the beginning, the authors mention the transition zone, and in most cases this concerns progressing urbanization.

I think that after some corrections the article is suitable for publication in Land.

Reviewer 3 Report

The authors accepted the comments.

Round 2

Reviewer 1 Report

The paper is revised well.

This manuscript is a resubmission of an earlier submission. The following is a list of the peer review reports and author responses from that submission.

Round 1

Reviewer 1 Report

The topic of the paper is very interesting and important. But the paper cannot be published because there are so many faults.

  1. Just one paragraph is for introduction. It is too rough for your journal. And I cannot understand the background and purpose of the study, at all. Please add the literature review section and provide the difference between existing studies and your paper.
  2. Methodology part is too brief. I cannot understand where the indicator come from. I feel that the methodology is just basic statistics with standardization.
  3. Where is a) and b) in figure 4? Please show the relation between figure 3 and figure 4. I cannot understand the figure 5 and 6 because there are so simple explanation. Please describe the meaning of the results based on your study area.
  4. I think the discussion section could be said without your result.
  5. I cannot read the summary, key finding, and limitation of the study in conclusion part.

Reviewer 2 Report

Article review: Analyze of Land-Use Change in Shortandy district in Terms of Sustainable Development.
The aim of study was an integrated assessment of the land use of the Shortandy district, which is the PUA of the metropolis of Nur Sultan, for sustainable development. The research objectives are the study of the spatiotemporal dynamics of the land use study area and the assessment of the level of sustainable development based on the currently available SDI.
Specific questions to consider in your evaluation include:
L 75: Methodology for Classifying land use, it's Accuracy Assessment and land use map generation is described in our previous work [30]. Methods of assessing sustainable development include four steps (Figure 2) [40] - The entire work methodology is based on 2 articles. The evaluated text lacks any description of the presented method. And this is required in scientific works (why these indicators etc.)
L115: The results of the calculation of SDI are shown in Table 4 (The individual SDI of Shortandy district in 2007-2017) ... there is no explanation for specific SDI indicators (where do these values, interpretation ...)
A very small number of land use classes - 5 Land use Classes
I understand that the lack of SDI analysis in 1992-2006 is due to the lack of data ...?
The discussion seems appropriate, but there is no practical reference to the purpose of the article.
Conclusions too obvious

The article is a fairly simple way to use statistical data (it is difficult to find a scientific method here). Lack of free space is a common problem in cities. Changes in land use in the transition zone described by the authors refer to arable land. Unlike the cases described in the literature (where agricultural use for urban development is disappearing in the transition zone). Describe why this is the case here.

It is difficult to say whether the example described is multidisciplinary. Spatial policy is conducted on different principles in different countries. This lack of brief planning characteristics in Kazakhstan. The article is even interesting, but it is difficult to say whether the described case study will find a wider audience. There is no description and possibilities to apply the procedure in other countries. For example, there are no forecasts and conclusions for the future (after 2018).

Reviewer 3 Report

Interesting article, but it needs to be completed. The authors combine land cover changes with SDI index development. Overall, however, I lack the outline of future developments in the discussion and in the conclusions. It is unclear whether current trends will be maintained or expected to change.
Some prediction of the country's development is missing. It is possible to use software for modeling, or eg. projects already approved to project into the current country and then describe their impact on the environment.
Similarly, the SDI index lacks a comparison with neighboring states respectively. discussion on future developments.
There are also formal errors in the text (for example, the sum of the total area values does not match after adding up each item).